# Point-of-Care Ultrasound (POCUS) in Pediatric Practice in Poland: Perceptions, Competency, and Barriers to Implementation—A National Cross-Sectional Survey

**DOI:** 10.3390/healthcare13151910

**Published:** 2025-08-05

**Authors:** Justyna Kiepuszewska, Małgorzata Gałązka-Sobotka

**Affiliations:** 1Department of Pediatrics, Florian Ceynowa Specialist Hospital, ul. Dr Alojzego Jagalskiego 10, 84-200 Wejherowo, Poland; 2Institute of Healthcare Management, Lazarski University, ul. Swieradowska 43, 02-662 Warsaw, Poland; m.galazka-sobotka@lazarski.edu.pl

**Keywords:** point-of-care ultrasound, pediatrics, POCUS, diagnostic imaging, medical education, cross-sectional study, Poland

## Abstract

**Background:** Point-of-care ultrasound (POCUS) is gaining recognition as a valuable diagnostic tool in various fields of medicine, including pediatrics. Its application at the point of care enables real-time clinical decision-making, which is particularly advantageous in pediatric settings. Although global interest in POCUS is growing, many European countries—including Poland—still lack formal training programs for POCUS at both the undergraduate and postgraduate levels. Nevertheless, the number of pediatricians incorporating POCUS into their daily clinical practice in Poland is increasing. However, the extent of its use and perceived value among pediatricians remains largely unknown. This study aimed to evaluate the current level of POCUS utilization in pediatric care in Poland, focusing on pediatricians’ self-assessed competencies, perceptions of its clinical utility, and key barriers to its implementation in daily practice. **Methods:** This cross-sectional study was conducted between July and August 2024 using an anonymous online survey distributed to pediatricians throughout Poland via national professional networks, with a response rate of 7.3%. Categorical variables were analyzed using the chi-square test of independence to assess the associations between key variables. Quantitative data were analyzed using descriptive statistics, and qualitative data from open-ended responses were subjected to a thematic analysis. **Results:** A total of 210 pediatricians responded. Among them, 149 (71%) reported access to ultrasound equipment at their workplace, and 89 (42.4%) reported having participated in some form of POCUS training. Only 46 respondents (21.9%) reported frequently using POCUS in their clinical routine. The self-assessed POCUS competence was rated as low or very low by 136 respondents (64.8%). While POCUS was generally perceived as a helpful tool in facilitating and accelerating clinical decisions, the main barriers to implementation were a lack of formal training and limited institutional support. **Conclusions:** Although POCUS is perceived as clinically valuable by the surveyed pediatricians in Poland, its routine use remains limited due to training and systemic barriers. Future efforts should prioritize the development of a validated, competency-based training framework and the implementation of a larger, representative national study to guide the structured integration of POCUS into pediatric care.

## 1. Introduction

Point-of-care ultrasound (POCUS) is transforming pediatric clinical practice by enabling real-time bedside diagnostics that facilitate faster and more accurate clinical decision-making [1].

Its dynamic, repeatable nature makes it particularly suited for pediatric patients, whose clinical status may change rapidly and who often present with nonspecific symptoms [2]. Moreover, POCUS reduces the reliance on ionizing imaging techniques, aligning with the ALARA (As Low as Reasonably Achievable) principle—a crucial consideration in children [3]. Another important factor is the shortage of pediatric radiologists in many healthcare systems. In response, the European Society of Paediatric Radiology (ESPR) has advocated for the use of POCUS by non-radiologists [4].

POCUS has a wide range of pediatric applications, supporting both diagnostic assessments—such as respiratory, abdominal, cardiac, and urinary tract conditions—and procedural guidance, including vascular access, bladder catheterization, and lumbar puncture [5,6,7,8,9]. Globally, the integration of POCUS into pediatric care is advancing, supported by an expanding body of evidence and increasing inclusion in formal medical education, especially in countries like the United States and Canada [10,11]. In contrast, Poland lacks standardized training pathways at both the undergraduate and postgraduate levels. In Poland, pediatricians’ increasing interest in learning POCUS is evidenced by the growing number of commercial courses offered. While the growing interest among Polish pediatricians is evident through the proliferation of private POCUS training courses [12,13,14,15,16,17,18], these remain voluntary, unregulated, and unstandardized. These courses are often costly, leading to unequal access and variability in training quality.

Beyond educational gaps, other commonly reported barriers to POCUS implementation include limited access to equipment, a lack of institutional support, the absence of national guidelines, limited time available for training, and concerns about medico-legal risk—all of which have been described in various international settings [19,20,21,22,23]. However, it remains unclear whether these challenges are equally relevant to the Polish pediatric context, as no national data currently exist on POCUS availability, usage, or perceived barriers.

The aim of this study was to examine the current use of POCUS in general pediatric clinical practice in Poland. Specifically, we sought to (1) assess the availability of POCUS equipment and training among pediatricians; (2) evaluate self-reported competencies and the frequency of POCUS use; and (3) identify key perceived benefits and barriers to its broader implementation.

To our knowledge, this is the first nationwide survey to explore the adoption of POCUS in Polish pediatric practice. In the absence of official data or national registries, this study addresses an important knowledge gap and may inform future strategies for structured implementation.

## 2. Materials and Methods

This study was designed as a nationwide, cross-sectional survey. Data were collected using an anonymous online questionnaire. The survey targeted general pediatricians from all 16 voivodeships in Poland, including both residents and specialists, working in hospital and outpatient settings. Subspecialists were excluded by design, as the survey focused on general pediatric practice.

A link to the survey was distributed via email to members of the Polish Pediatric Society (PTP), which includes approximately 2873 individuals. The survey remained open from July to August 2024. Out of all of the recipients, 1039 opened and viewed the questionnaire (engagement rate: 36%), and 210 completed the survey in full. The final response rate—defined as completed questionnaires included in the main analysis—was 7.3%.

The study was reviewed by the Bioethics Committee of the Regional Medical Chamber in Gdansk, which issued a formal statement (No. KB-3a/25) confirming that this study does not require ethical approval due to its anonymous and non-interventional nature.

The questionnaire was developed by the lead author, a certified pediatric ultrasonographer with hands-on experience in POCUS implementation and training. Its content was based on clinical insights and adapted from prior POCUS-related surveys published in the literature [23]. The survey was designed to explore key themes, including the availability of equipment and training courses, POCUS usage rate in daily practice, self-assessed competency, perceived educational needs, and implementation barriers. The survey included Yes/No questions, 5-point Likert or quasi-Likert scale questions, ordinal closed-ended questions, multiple-choice options, and open-text “Other” fields. Several questions used Likert-type or simplified categorical response formats (e.g., “Very high,” “High,” “I don’t know,” “Low,” “Very low”) and qualitative frequency labels such as “rarely,” “often,” or “very often.” These were kept in their original textual form.

In order to ensure content validity, a draft version of the questionnaire was piloted with ten pediatricians who did not participate in the main study. Based on their feedback, adjustments were made to improve question clarity, terminology, and layout. The final version was also reviewed for content accuracy by the second author of this study, an experienced expert in the implementation of innovative technologies within the Polish healthcare system. The full version of the questionnaire is provided in the Appendix A.

The respondents answered the survey based on their individual experience with POCUS in their everyday clinical work and shared their opinions on barriers to its implementation. Participation was voluntary and anonymous. No identifying data, such as names, email addresses, or IP addresses, were collected. The survey was conducted using Google Forms with anonymity settings enabled. Upon closing the survey, data were downloaded and stored in password-protected files accessible only to the research team. All data-handling procedures complied with the General Data Protection Regulation (GDPR).

The data analysis included the following:Descriptive statistics to summarize the distribution of quantitative variables using basic descriptive measures.Chi-square tests of independence were applied to analyze associations between categorical variables (e.g., training participation, frequency of use, competency, equipment access). Effect sizes were reported using Cramér’s V.Spearman’s rho was used to evaluate correlations between ordinal or continuous variables (e.g., work experience and POCUS use).Frequency analysis to present the percentage distributions of nominal and ordinal variables.Missing data were minimal; cases with missing responses were excluded from respective analyses. No imputation techniques were used.Open-ended responses in the survey were analyzed using deductive thematic content analysis. A predefined set of thematic categories (codebook), based on the objectives of the study and the related literature, was used to guide the analysis. Two authors independently coded the responses manually, with final categorization agreed upon by consensus. Each summarized statement presented in the results represents a cluster of thematically similar responses. No automated coding software was used in this process.The statistical analysis was conducted using the Jamovi software, version 2.3.24. package, and Microsoft Excel.

## 3. Results

A total of N = 210 individuals participated in the survey. The majority of the respondents were female (86.2%), and the group was divided fairly evenly between pediatric specialists (56.0%) and residents (44.0%). Most participants worked in a hospital setting, either exclusively (52.6%) or in combination with providing outpatient care (25.8%). In terms of age, nearly half of the respondents were under 34 years of age (46.4%). The highest number of responses came from the Pomorskie (32.4%) and Mazowieckie (20.0%) voivodeships, followed by Śląskie and Wielkopolskie. The demographic and professional characteristics of the respondents are summarized in Table 1.

The mean length of work experience among the participants was M = 10.29 years (SD = 9.279), with a median of 8.00 years. The minimum reported experience was 0.5 years, and the maximum was 54.0 years, indicating a wide range of clinical seniority. The distribution was positively skewed (skewness = 2.01), with a high kurtosis value (5.03), suggesting a concentration of responses in the lower range with a few outliers reporting very high levels of experience. Descriptive statistics regarding the years of professional experience are presented in Table 2.

Using chi-square and correlation tests, we examined how training, equipment availability, perceived competencies, age, and professional experience influenced the frequency of POCUS use and attitudes toward its clinical utility. Detailed statistical associations between the selected variables and frequency of POCUS use are presented in Table 3.

The table presents the results of the chi-square tests and Spearman’s rank correlation examining associations between the selected variables and the frequency of POCUS use. Statistically significant relationships were found between POCUS use and participation in training, equipment availability, self-assessed competency, age, and perceived clinical need. No significant associations were observed between training participation and the perceived impact on diagnostics or professional prestige. The effect sizes (Cramér’s V) ranged from small to strong. Specifically, V = 0.50 (equipment vs. use) indicated a large effect, V = 0.44 (training vs. use) indicated a moderate effect, and V = 0.24 (age vs. training) indicated a small-to-moderate effect. Effect sizes below 0.2 were considered weak and should be interpreted with caution. *p* < 0.05 was considered statistically significant. Although no formal correction for multiple comparisons was applied due to the exploratory nature of the study, effect sizes were reported to aid in the interpretation of the clinical relevance.

The distribution of the responses was further examined using descriptive statistics. Most of the participants, 71% (n = 149), reported that POCUS equipment was available at their place of work (Figure 1a). Regarding the implementation of POCUS in clinical practice, only 21.9% (n = 46) of the respondents reported using the method frequently in their daily routine, and almost half of the respondents, 46,2% (n = 97), reported lacking the necessary skills to perform POCUS in daily clinical practice (Figure 1b). The remaining participants indicated that they perform POCUS in their routine work either “rarely” (n = 37; 17.6%) or “occasionally” (n = 30; 14.3%) (Figure 1b).

Almost half of the respondents, 42.4% (n = 89), reported having participated in POCUS training courses, while an additional 21.0% (n = 44) indicated plans to attend such training in the future (Figure 2a). Among them, 32.9% (n = 69) believed that there are sufficient training opportunities for pediatric POCUS, although high admission costs were frequently cited as a limiting factor (Figure 2b).

Detailed results regarding equipment availability, the frequency of POCUS use in daily clinical practice, and participation in training are presented in Figure 1 and Figure 2. Despite relatively good access to ultrasound equipment and participation in POCUS training courses, only a portion of pediatricians continue to use the method in daily clinical practice. This suggests a need for more practical, hands-on training, ongoing mentorship, and system-level support to bridge the gap between theoretical knowledge and routine clinical application.

The respondents were asked to self-assess their competency in performing POCUS (Figure 3a). The majority rated their skills as “very low” (n = 72; 34.3%) or “low” (n = 64; 30.5%), and one in five participants (n = 42; 20.0%) admitted they were unsure about their skill level. Only a minority assessed their competency as “high” (n = 29; 13.8%) or “very high” (n = 3; 1.4%). These results indicate that most pediatricians feel inadequately trained in POCUS techniques. In contrast, the perceived clinical demand for POCUS (Figure 3b) is markedly high (n = 97; 46.2%) or very high (n = 72; 34.3%). This disparity between the perceived clinical need and self-assessed readiness underscores a critical gap in training and preparedness among pediatricians in Poland.

The respondents also evaluated the clinical benefits of implementing POCUS examinations into daily practice (Figure 4). The respondents were allowed to select more than one response. The data are presented as percentages of the total respondents (n = 210). The vast majority indicated that POCUS facilitates clinical decision-making (n = 199; 94.8%) and reduces the time to diagnosis (n = 189; 90%). Additionally, 78.6% (n = 165) reported that it decreases the need for referrals for invasive tests (e.g., X-ray or CT), and 71.9% (n = 151) believed it improves the diagnostic accuracy. Only a small percentage felt that POCUS makes work more difficult due to the time involved (n = 18; 8.6%) or brings no benefits at all (n = 1; 0.5%).

In addition to providing structured responses to questions, the participants were invited to share their opinions on other perceived advantages of POCUS through open-ended questions. The qualitative content analysis revealed several recurring themes, which are summarized in Appendix A. The respondents highlighted that POCUS in primary settings may reduce unnecessary referrals and diagnostic delays, contributing to more efficient healthcare delivery. Cost-related benefits were also noted, including fewer imaging and laboratory tests, reduced antibiotic use, and lower public healthcare expenses. Clinically, POCUS was seen as a valuable tool for differentiating infectious diseases and guiding targeted treatment, especially in infants and non-cooperative patients. Several respondents also mentioned improved communication with caregivers, increased trust in the diagnosis, and reduced patient stress.

In terms of professional development and perception, the majority of the respondents acknowledged the positive influence of POCUS on their work satisfaction and the perceived prestige of the pediatric profession (Figure 5a). When asked “Do you think that acquiring POCUS skills and enhancing your professional qualifications would have a positive impact on your job satisfaction?”, 86.6% of the participants (n = 181) answered “Yes”, while 9.6% (n = 20) had no opinion, and 3.8% (n = 8) answered negatively. In a separate question assessing the potential of POCUS to elevate the status of pediatricians, 38.6% (n = 81) of the respondents selected “major impact”, and 29.5% (n = 62) chose “significant impact.” These findings suggest that over two-thirds of participants believe that the implementation of POCUS could meaningfully contribute to the professional prestige of pediatricians.

Most of the respondents recognized the potential of POCUS to improve healthcare system efficiency (Figure 5b). In response to the question “Do you think that POCUS can shorten the diagnostic pathway in pediatric patients?”, 52.9% of the participants (n = 111) indicated a major impact, while 33.8% (n = 71) reported a significant impact. Collectively, these responses represent 86.7% (n = 182) of the study group, reflecting a prevailing view that POCUS can meaningfully accelerate the diagnostic process in pediatric care. Similarly, when asked “Do you believe that using POCUS in pediatrics can lead to cost savings in the Polish healthcare system?”, the majority of the respondents (73.3%, n = 154) answered “Yes”, suggesting a strong belief in the economic value of POCUS. Meanwhile, 16.7% (n = 35) had no opinion, and 10% (n = 21) responded negatively. These findings indicate that the integration of POCUS into pediatric practice is perceived not only as clinically effective but also as beneficial at the system level, with the potential to optimize resource utilization and reduce expenditures.

The respondents identified a lack of support from public institutions (n = 165; 78.6%) and a lack of appropriate training (n = 150; 71.4%) as the main barriers to implementing POCUS in everyday pediatric practice in the Polish healthcare system (Figure 6).

The respondents also reported additional barriers to performing POCUS in clinical practice (expressed through open-ended questions), as presented in Appendix A. The open-ended responses revealed additional barriers not fully captured in the closed-ended questions, such as a lack of time for training and exam completion (especially in primary care), generational resistance to new technologies, and limited trust in POCUS findings by senior physicians. Cultural perceptions—such as the belief that ultrasound is the exclusive domain of radiologists—and insufficient leadership support were also frequently mentioned.

Figure 7 presents the respondents’ answers to a multiple-choice question regarding key factors that support the implementation of POCUS in daily pediatric practice. Overall, 69% of the physicians (n = 145) indicated that introducing mandatory POCUS training into specialization programs or medical education is a key factor. The distribution of the remaining responses is presented below.

The responses to the open-ended question regarding strategies to overcome barriers to POCUS implementation were analyzed thematically. A summary of the thematic categories and representative quotes is presented in Appendix A. The open-ended responses on strategies to improve POCUS implementation emphasized the need for systemic solutions such as structured funding, the integration of ultrasound into the pediatric specialization curriculum, and greater support from professional societies. The respondents also highlighted practical needs like better access to equipment and protected time for examinations and training.

## 4. Discussion

To our knowledge, this is the first nationwide exploratory study investigating the use and implementation of point-of-care ultrasound (POCUS) in general pediatric practice in Poland. While this study is limited by a low response rate and cannot claim full representativeness of the national pediatric population—estimated at approximately 15,501 pediatricians and 2077 residents [24,25]—it includes responses from all administrative regions of Poland, encompassing various healthcare settings and career levels. This diversity lends value to the data and provides a preliminary yet informative overview of the current landscape of POCUS use. This study sheds light on the patterns of access, perceived clinical relevance, training needs, and practical barriers, offering useful directions for further research and system-level reflection.

Despite the relatively high reported access to ultrasound equipment (71%) and growing awareness of POCUS’s clinical value, our findings suggest that its routine use in pediatric practice remains limited: only 21.9% of the respondents reported frequent use. At the same time, 80.5% indicated a high or very high perceived clinical demand for POCUS, and 86.6% believed that acquiring related skills could enhance their professional satisfaction. These findings—combined with the existing literature highlighting POCUS’s role in pediatric practice [26,27,28,29,30,31,32,33]—suggest interest and perceived utility among clinicians, warranting further exploration of barriers and opportunities for safe and effective implementation in the Polish context.

Our findings suggest that the availability of equipment, a growing commercial training market, and strong motivation among pediatricians may not be sufficient to ensure the widespread adoption of POCUS in routine practice. This mismatch between access and interest on the one hand, and underutilization on the other, highlights the need to explore underlying barriers.

Our statistical analysis supports the existence of this discrepancy by demonstrating that participation in POCUS training, access to ultrasound equipment, and younger age were all significantly associated with more frequent POCUS use and higher self-assessed competency. The pediatricians who had completed training were several times more likely to report regularly using POCUS compared to those without training. Similarly, access to equipment emerged as a critical enabling factor, with frequent use reported by 30% of clinicians with access, versus only 3% among those without. Interestingly, neither training nor clinical experience significantly influenced the perceptions regarding POCUS’s impact on diagnostic efficiency or professional prestige. These findings highlight that while technical and educational factors are necessary, they are not sufficient to shift deeper clinical attitudes or institutional behaviors, further underscoring the need for a systemic implementation strategy.

This pattern is not unique to Poland and reflects broader international challenges in translating POCUS’s use into routine clinical use. For example, in the United States, 96% of pediatric emergency departments had ultrasound devices, yet only 61% reported regular use [34]. Similarly, Musolino et al. [35] found that although 87.1% of Italian pediatric emergency and inpatient settings had access to ultrasound, a substantial proportion (44.3%) of residents did not use it. In a European–Israeli survey, Parri et al. [36] observed that training was relatively common (61.9%), but clinical application remained limited. These findings suggest that access and training alone may be insufficient to ensure sustained implementation.

A key barrier identified in our study was the lack of structured training, reported by 71.4% of the respondents, along with insufficient institutional support (78.6%) and absence of official guidelines from scientific societies (54.3%). Notably, although 42.4% had completed a POCUS course, nearly two-thirds still rated their competency as low or very low.

Although the growing number of commercial POCUS courses in Poland reflects increasing interest, the educational landscape remains fragmented, unstandardized, and reliant on individual initiative. Training is typically self-funded and of a short duration (Appendix A), which may limit long-term competency. Evidence suggests that without ongoing clinical practice, supervision, and feedback, skills deteriorate over time [37,38,39,40].

According to current evidence, no country—including Poland—has yet integrated mandatory POCUS training into general pediatric residency curricula. While Polish pediatricians currently rely on self-funded and non-standardized training, several high-income countries have introduced more structured educational pathways, particularly at the subspecialty level. In the United States, formal POCUS training is primarily integrated into pediatric subspecialty fellowships, such as pediatric emergency medicine and anesthesiology, with support from national networks and accreditation bodies [41,42]. In Canada, selected universities mandate POCUS training as part of PEM fellowships, including supervised practice and assessment [43,44,45]. Professional bodies such as the AAP, ABP, and ACGME have acknowledged the clinical value of POCUS and integrated it into pediatric emergency medicine standards [46,47,48]. These examples highlight possible models for formalizing training and ensuring skill retention.

These international efforts to improve POCUS training and infrastructure directly resonate with our findings, which highlight the lack of national guidelines and structured education as key barriers in Poland. For instance, while the European Society for Emergency Medicine (EUSEM) advocates for standardized POCUS competence in pediatric care [49], no similar framework currently exists in Poland. Likewise, the institutional support observed in UK pediatric units [50] contrasts with the limited infrastructure and institutional engagement reported by our respondents.

Despite international efforts to promote POCUS education and use, our findings suggest that in Poland, the absence of a national training framework, formal accreditation, and institutional support contributes to its limited clinical adoption. Although educational gaps are central, they alone do not explain the underuse observed. The respondents also cited additional barriers—including time constraints, unclear medico-legal responsibilities, a lack of mentorship, and cultural resistance among senior staff—which likely compound the implementation challenge. These factors, identified both quantitatively and qualitatively in our study, warrant further attention in future interventions.

In many Polish institutions, diagnostic imaging is traditionally reserved for radiologists, which may discourage pediatricians from independently using POCUS—even after completing training. This cultural barrier, reported by the respondents in our study, reflects broader trends also observed internationally, particularly in general pediatrics outside of subspecialties such as emergency medicine [51].

To contextualize the observed barriers and inform future strategies, we mapped our findings onto the Consolidated Framework for Implementation Research (CFIR). The reported obstacles aligned with all five CFIR domains [52], reflecting the multilevel complexity of implementing POCUS in pediatric care. At the intervention level, the lack of standardized, affordable training limits equitable access. At the outer setting level, systemic inertia is reinforced by the absence of national guidelines, funding mechanisms, and policy incentives. Inner setting barriers include insufficient infrastructure, administrative support, and protected time. At the individual level, many of the respondents reported low confidence and uncertainty about their competence. Finally, the lack of a structured implementation process—such as mentorship, supervision, or a strategic rollout—further hampers clinical integration.

This framework supports a coordinated, multilevel response to address these interrelated challenges and promote sustainable POCUS adoption in pediatric practice in Poland.

While these findings highlight important directions for future development, several limitations of our study should be considered when interpreting the results.

Firstly, the overall response rate was low. According to data obtained from the Polish Pediatric Society (PTP), the survey was emailed to 2873 members with active email addresses in the database. Of these, 1039 recipients opened the message, and 210 completed the questionnaire, yielding an estimated response rate of approximately 7.3% among those who opened the email. This low response rate may introduce systematic bias and limit the generalizability of the findings. In particular, non-response bias is possible, as individuals who are more interested in POCUS or who work in regions with more active educational and clinical champions (e.g., the Pomorskie voivodeship) may have been more likely to participate. As a result, the data may overrepresent more motivated and better-equipped environments, thus inflating reported access or usage levels.

Secondly, the study was exploratory in nature, and no formal power calculation was performed prior to data collection. As a result, the sample size was not determined based on statistical assumptions, and the findings should be interpreted as hypothesis-generating rather than confirmatory.

Thirdly, although the questionnaire was adapted from previous surveys and reviewed by clinical experts, it was not formally validated using psychometric methods. In addition, some response options included qualitative frequency labels such as “rarely,” “often,” or “very often” that were not numerically defined, which may have introduced inconsistency in interpretation across the respondents.

Fourthly, this study relied entirely on self-reported data, without objective measures of POCUS use, competency, or patient outcomes. This increases the risk of social desirability bias, where participants may overestimate their skills or frequency of use in order to align with perceived professional expectations or norms.

Fifthly, the survey was distributed using Google Forms, and although anonymity settings were enabled, the system did not prevent multiple submissions from the same individual, nor did it allow for the geographic response distribution to be tracked in detail. The lack of duplicate response control is a recognized technical limitation of many online survey platforms.

Finally, the study was not preregistered in a formal research registry. While this does not affect the integrity of the data, preregistration improves transparency by specifying study objectives, hypotheses, and analysis plans in advance. In future work, we plan to use platforms such as the Open Science Framework (OSF) to register similar protocols. Nonetheless, we confirm that the design and reporting of the study were guided by the STROBE (Strengthening the Reporting of Observational Studies in Epidemiology) and CHERRIES (Checklist for Reporting Results of Internet E-Surveys) checklists, where applicable.

Despite these limitations, this study is among the first in Poland to analyze the use of point-of-care ultrasound among pediatricians. Its strengths include its national scope, inclusion of both residents and board-certified pediatricians, and representation of hospital and outpatient care settings.

## 5. Conclusions

Although POCUS is perceived as clinically valuable by the surveyed pediatricians in Poland, its routine use remains limited due to training and systemic barriers. Future efforts should prioritize the development of a validated, competency-based training framework and the implementation of a larger, representative national study to guide the structured integration of POCUS into pediatric care.

## Figures and Tables

**Figure 1 healthcare-13-01910-f001:**
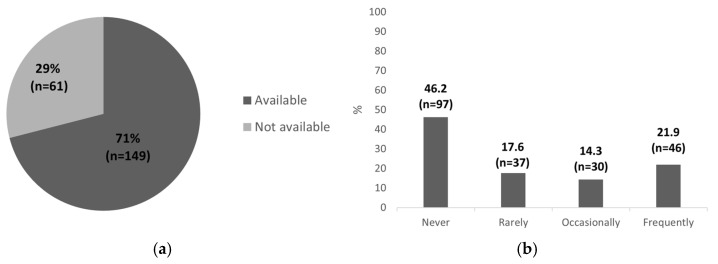
(**a**) Availability of POCUS equipment; (**b**) frequency of POCUS use. Data are expressed as a percentage with absolute sample sizes (n = 210).

**Figure 2 healthcare-13-01910-f002:**
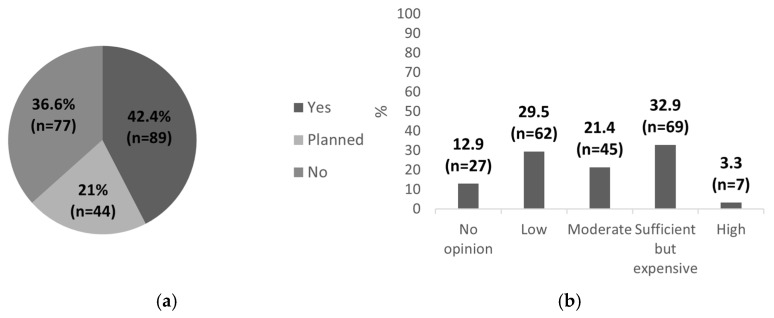
POCUS training in Poland. (**a**) Participation in training courses; (**b**) perceived availability of training opportunities. Data shown as percentages with absolute sample sizes (n = 210).

**Figure 3 healthcare-13-01910-f003:**
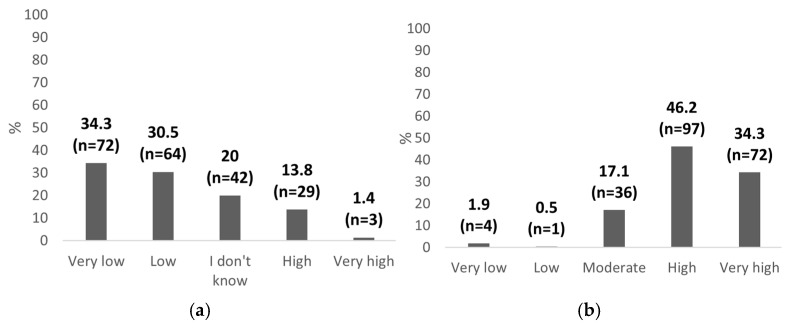
(**a**) Self-assessed POCUS competency; (**b**) perceived demand for POCUS in pediatrics. Data shown as percentages with absolute sample sizes (n = 210).

**Figure 4 healthcare-13-01910-f004:**
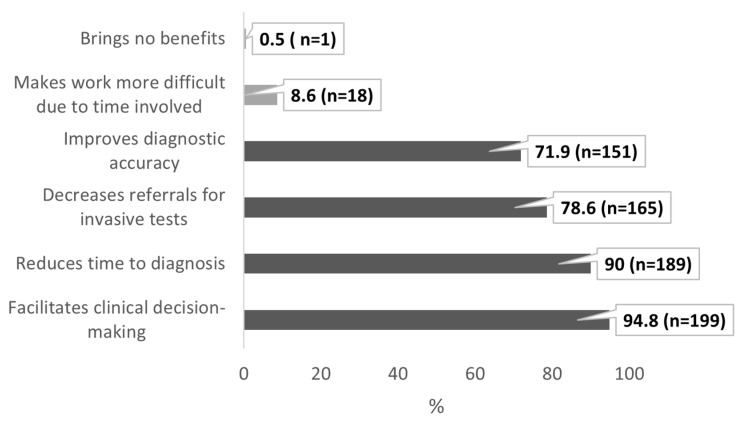
Perceived benefits and barriers of POCUS use in pediatric practice (multiple answers allowed). Data are expressed as a percentage with absolute sample sizes (n = 210).

**Figure 5 healthcare-13-01910-f005:**
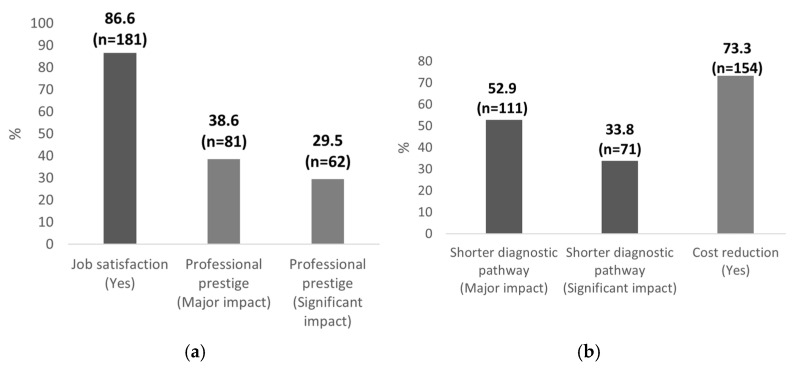
Positive responses regarding the benefits of POCUS: (**a**) professional benefits; (**b**) systemic benefits. Data are expressed as a percentage with absolute sample sizes (n = 210).

**Figure 6 healthcare-13-01910-f006:**
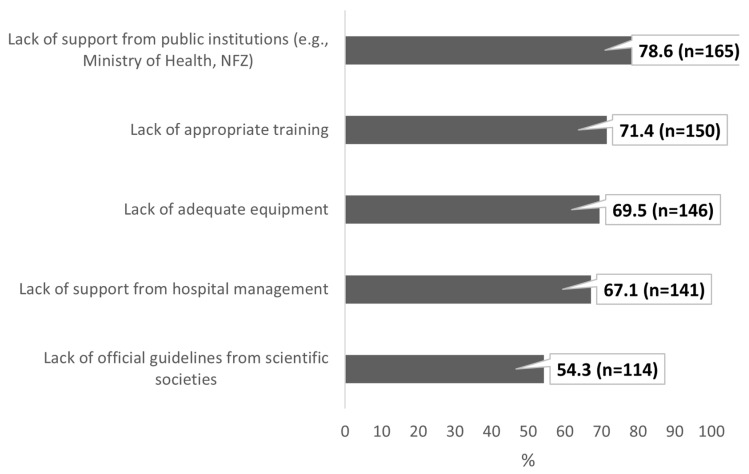
Reported barriers to POCUS implementation in pediatric practice (multiple responses allowed). Data are expressed as a percentage with absolute sample sizes (n = 210).

**Figure 7 healthcare-13-01910-f007:**
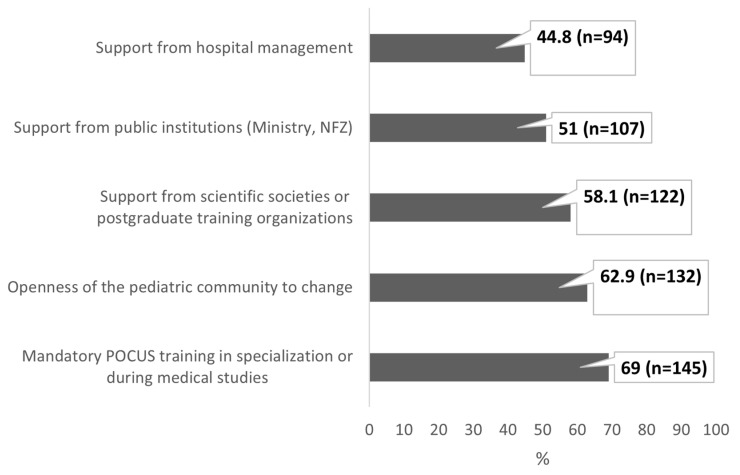
Factors supporting the implementation of POCUS in daily practice (multiple-choice questions). Data are expressed as percentages with absolute sample sizes (n = 210).

**Table 1 healthcare-13-01910-t001:** Basic characteristics of respondents.

Variable	N	n (%)
Gender	210	
Female		181 (86.2%)
Male		29 (13.8%)
Professional status	209	
Pediatric specialist		117 (56.0%)
Pediatric resident		92 (44.0%)
Employment sector	209	
Hospital care		110 (52.6%)
Hospital and outpatient care		54 (25.8%)
Outpatient care only		45 (21.5%)
Age group	209	
Under 34 years		97 (46.4%)
35–44 years		84 (40.2%)
45–59 years		20 (9.6%)
60 years and older		8 (3.8%)
Region of practice (voivodeship)	210	
Pomorskie		68 (32.4%)
Mazowieckie		42 (20.0%)
Śląskie		23 (11.0%)
Wielkopolskie		13 (6.2%)
Kujawsko-Pomorskie		11 (5.2%)
Małopolskie		10 (4.8%)
Dolnośląskie		8 (3.8%)
Podkarpackie		8 (3.8%)
Zachodniopomorskie		6 (2.9%)
Łódzkie		5 (2.4%)
Lubuskie		3 (1.4%)
Podlaskie		3 (1.4%)
Świętokrzyskie		3 (1.4%)
Lubelskie		2 (1.0%)
Opolskie		2 (1.0%)
Warmińsko-Mazurskie		2 (1.0%)
Lubelskie, Mazowieckie *		1 (0.5%)

* One respondent reported working in both the Lubelskie and Mazowieckie voivodeships; therefore, this dual-region response is listed as a separate combined category.

**Table 2 healthcare-13-01910-t002:** Descriptive statistics for work experience.

N	M	SD	Me	Min	Max	Skewness	Kurtosis
210	10.29	9.279	8.00	0.5	54.0	2.01	5.03

Note: N = sample size; M = mean; SD = standard deviation; Me = median; Min = minimum; Max = maximum.

**Table 3 healthcare-13-01910-t003:** Associations between selected factors and the frequency of POCUS use among pediatricians.

Variable	Chi-Square/Spearman’s ρ	*p*-Value	Cramér’s V	Notable Results
Participation in training	χ^2^(6) = 80.07	<0.001	0.44	Proportion of 38% trained vs. 8% untrained participants use POCUS frequently
Self-assessed competency	χ^2^(8) = 75.14	<0.001	0.42	Higher ratings among trained participants
Equipment availability	χ^2^(6) = 80.07	<0.001	0.50	Proportion of 30% of participants with access use frequently vs. 3% without
Years of experience	ρ = −0.12	0.080	–	Negative trend, not significant
Age vs. training participation	χ^2^(6) = 23.39	<0.001	0.24	Younger physicians more likely to be trained
Perceived need for POCUS	χ^2^(6) = 17.00	0.030	0.20	Trained physicians more likely to perceive high clinical need
Diagnostic impact perception	χ^2^(6) = 8.20	0.414	0.14	No significant difference
Professional prestige perception	χ^2^(6) = 7.23	0.512	0.13	No significant difference

Note: Chi-square statistics (χ^2^) and Spearman’s rank correlation (ρ) were used to test associations. Cramér’s V indicates effect size; *p* < 0.05 was considered statistically significant.

## Data Availability

The original contributions presented in this study are included in the article/Appendix A. Further inquiries can be directed to the corresponding author.

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
