# Peer review of "Point-of-Care Ultrasound (POCUS) in Pediatric Practice in Poland: Perceptions, Competency, and Barriers to Implementation—A National Cross-Sectional Survey"

_healthcare, 2025, doi:10.3390/healthcare13151910_

Round 1
Reviewer 1 Report
Comments and Suggestions for Authors
Major Revision
The manuscript presents a timely and well-structured national cross-sectional survey exploring the implementation of point-of-care ultrasound (POCUS) in pediatric practice in Poland. The study addresses an important and under-researched area, and the data are clearly presented. There are a few points that should be addressed to strengthen the manuscript further:
1.Categorize the types and applications of POCUS more clearly.
2.Although the authors point out that the non-interventional design eliminated the need for ethical approval, it would be beneficial to provide certain Polish laws or ethical standards that support this exemption. To promote openness, the authors should also think about requesting institutional explanation or retroactive ethical approval.
3.Descriptive statistics are being used in the analysis. Although this offers a helpful summary, the inclusion of fundamental inferential tests (such as chi-square, t-tests, or correlations) might aid in analyzing the connections between important factors like clinical experience and POCUS utilization.
4.Although the high cost of training is mentioned in the manuscript as a barrier, it would be helpful to briefly compare the price of POCUS training in Poland with that of other nations. Expand the discussion of barriers using internationally accepted frameworks.
5."Lubelskie" appears twice in Table 1 with distinct values. Please specify if this represents a particular subgroup, a data input error, or a geographical overlap.
6.The text mentions that the survey instrument is included in the Appendix, however the version under review does not have the appendix. This information is crucial for repeatability and transparency, so please make sure it is included in the final submission.
7.Figures are labeled (e.g., “(a)” and “(b)”), ensure these labels are consistently and clearly referenced within the main text.
8.it would benefit from the inclusion of more recent systematic reviews or meta-analyses on POCUS implementation strategies in pediatrics or similar healthcare settings,
9.Consider brief commentary on how findings might guide policy recommendations or curriculum development
Author Response
Thank you for your helpful comment. We have carefully revised the manuscript to improve clarity, grammar, and overall English language quality. The revised version has been thoroughly proofread to ensure that the research aims, methods, and findings are now presented in a clearer and more professional manner.

Reviewer 2 Report
Comments and Suggestions for Authors
Thank you for allowing me to review. The manuscript is well-done and easy for readers to follow. There are some minor improvements for the authors to consider:
- Introduction: Can you provide an estimated number of pediatric specialists and residents in Poland to help understand if the response rate is representative? Do all pediatric specialists and residents automatically become members of the society that was used to deploy the survey or did it only going to paying members?
- Methods should include the process of thematic review (ie was it completed by 1 individual or more? was there a review and consensus process or codebook?)
- Results: The pie charts are not typical for manuscripts, but it was easy to follow from a reader standpoint for the most part. A table would allow for p-values to be displayed and to see more results all together. a) Can you add a subgroup analysis of those who were the "never" respondents for using POCUS in practice, the ones who did not have equip available? What is the subset of clinicians who have the technology, but are not using it? Does the practice setting matter? b) Were there any definitions to frequency or definition of what types of equipment count as POCUS? c) Double check consistency of figure/table subtext [ie, Figure 4 and a few others explain the question context (this is not usual presentation, I would recommend removing and adding the the body text) "The questions were multiple choice, and the percentage distribution of responses is shown."] d) For each table with qualitative analysis, are these rank orders of highest frequency of responses? should give # of respondents who cited each theme. e) Figure 6 percentage does not match text, please correct and verify all percentages. f) All figures should provide and N of total number of respondents as responders can drop off during a survey or choose to not answer specific questions. g) Figure 7 does not say if respondents picked the main barrier (only select 1) or barriers (able to select more than 1). Providing specifications of N and how the questions was asked would improve this.
- Limitation that the survey was not validated before use. It was reviewed and edited, but I don't see mention of validation statistics done on the pilot survey. Are there payment issues with POCUS since it doesn't appear in most clinical guidelines? I don't see that listed in the survey or limitations, but in the U.S. payment is a huge factor in what tests get run.
Author Response
Thank you for your detailed and constructive feedback. We appreciate your thorough review, which has helped us improve the quality of our manuscript.
All point-by-point responses to your comments, along with detailed explanations of the revisions made, have been included in the attached Word document.
We kindly invite you to refer to the file below for our full responses.

Reviewer 3 Report
Comments and Suggestions for Authors
This manuscript addresses an important topic. The nationwide survey offers preliminary insights into access, perceived utility, and barriers to implementation. However, it lacks analytical depth, methodological transparency, and critical discussion. Major revisions are needed. Below are my comments:
- The abstract lacks sufficient quantitative depth and transparency. While it reports percentages, it fails to provide denominators and comparative context. It does not address a key methodological limitation: the inability to calculate a response rate, which critically impacts the generalizability of the findings. The abstract should acknowledge these limitations and clarify that the analysis is currently descriptive only.
- The introduction is overly descriptive and does not highlight the specific research gap within Poland. It lacks a clear set of study objectives at the end to guide the reader. It should explicitly link the need for this study to the absence of national data on the adoption of POCUS among pediatricians in Poland.
- The methods section requires expansion to ensure transparency and reproducibility. The manuscript does not indicate how many pediatricians were contacted, which prevents the calculation of a response rate. It also fails to address whether the sample is representative by region, practice type, or subspecialty. Moreover, the survey instrument is not included, and the coding and handling of Likert scale data are not described. The thematic analysis lacks methodological detail, including coder independence, validation processes, and the software used. Notably, the study does not include inferential statistical analyses to explore potential associations between variables such as training participation, years of experience, and frequency of POCUS use.
- The results section is limited to descriptive reporting without inferential analysis, which constrains the interpretation and value of the findings. There are numerous figures, some of which are redundant. The results of the thematic analysis are reported without illustrative quotes, reducing the depth and transparency of the qualitative findings. Furthermore, there is no stratification of results by relevant variables such as age, gender, region, or professional status, and the potential relationship between self-assessed competence and training participation remains unexplored.
- The discussion reiterates descriptive findings without critically examining the underlying reasons for the low uptake of POCUS, despite the availability of equipment and clinician interest. There is minimal exploration of systemic, cultural, financial, or institutional barriers specific to Poland, and the interplay between training, equipment availability, and institutional support is not addressed. The discussion should critically analyse these factors and propose actionable recommendations for addressing barriers to POCUS implementation, drawing on successful frameworks from other countries where relevant. It also fails to acknowledge potential study limitations, such as selection bias and regional imbalances in responses.
- The conclusion is repetitive and lacks forward-looking recommendations for research, policy, or training. It should outline clear next steps to integrate POCUS into pediatric practice in Poland further, focusing on the need for structured training, guideline development, and institutional policy initiatives.
- Figures and tables require improved clarity and consistency. Redundant figures should be consolidated, and a consistent colour palette with clear labelling should be used across all visuals. Tables summarising qualitative themes should include direct quotes to enhance the credibility of the thematic analysis.
- Finally, the ethical considerations and data management statements are brief and should be expanded. The manuscript should reference relevant Polish guidelines to justify the absence of ethics committee approval and explicitly state how participant confidentiality was ensured, as well as how data were securely stored and managed.
Author Response

(The authors gave the same response as above.)

Round 2
Reviewer 3 Report
Comments and Suggestions for Authors
The manuscript provides the first comprehensive overview of point-of-care ultrasound (POCUS) usage among Polish pediatricians. However, it requires improvements in methodological transparency, statistical rigor, and cautious interpretation. The following are my observations:
1. While the introduction offers a broad literature context, it meanders into mini-reviews of POCUS applications and global curricula, diluting the central argument. This section could be more focused on the unresolved issue of national uptake in Poland, should cite only the most pertinent evidence, and end with a clear statement of objectives and hypotheses.
2. The national scope, ethical approval, and mixed-methods approach are strengths, but crucial details are either weak or missing. The sampling frame is limited to email lists from the Polish Pediatric Society, and the final completion rate is just 7.3%, with no power calculation to support its adequacy. The custom survey instrument is not psychometrically validated, and the study was neither preregistered nor reported using CHERRIES or STROBE guidelines. The subjective response-frequency labels (“rarely,” “often”) should be defined numerically. These shortcomings underscore the need for a validation study, clearer denominator reporting, and adherence to best practice guidelines.
4. The results are well supported by graphics, but they depend heavily on raw percentages and inconsistent statistical reporting. Chi-square tests and correlation coefficients are provided without multiple comparison corrections, and effect sizes or confidence intervals are often missing. Some pie charts are redundant and could be combined, while a concise table of key quantitative findings paired with thematic codes and illustrative quotes would enhance clarity.
5. The discussion overextends by dedicating too much space to a global curriculum review rather than interpreting the new data. The assertion that the results “highlight a timely opportunity for the Polish healthcare system” seems overstated given the limited representativeness of the sample. A more balanced tone that highlights the exploratory nature of the work and reduces extraneous literature would reinforce credibility.
6. Although most limitations are mentioned, they are scattered throughout the manuscript. These should be consolidated into a single paragraph, expanding on issues such as non-response bias (since respondents are likely from regions with active POCUS champions), the lack of objective competency measures, and the potential for social-desirability bias in self-assessments.
7. The conclusion currently reiterates the results without providing additional insights. It should be condensed to one sentence that highlights the disparity between the high perceived value and low uptake of POCUS, followed by another sentence outlining concrete next steps, such as developing a validated training framework and conducting a larger, representative survey.
8. All charts lack absolute sample sizes and error bars. Some figures could be combined into a single multi-panel graphic, and extensive qualitative tables might be better relegated to supplementary material for a more streamlined presentation.
Comments on the Quality of English LanguageThe writing is verbose and includes inconsistent tense usage.
Author Response
"Please see the attachment"
